# Targeted Analysis of Sphingolipids in Turkeys Fed Fusariotoxins: First Evidence of Key Changes That Could Help Explain Their Relative Resistance to Fumonisin Toxicity

**DOI:** 10.3390/ijms23052512

**Published:** 2022-02-24

**Authors:** Philippe Guerre, Angelique Travel, Didier Tardieu

**Affiliations:** 1National Veterinary School of Toulouse, ENVT, Université de Toulouse, F-31076 Toulouse, France; 2ITAVI, L’Orfrasière, F-37380 Nouzilly, France; travel@itavi.asso.fr

**Keywords:** fumonisin, turkey, ceramide, sphingomyelin, dihydroceramide, deoxysphinganine, glucosylceramide, lactosylceramide

## Abstract

The effects of fumonisins on sphingolipids in turkeys are unknown, except for the increased sphinganine to sphingosine ratio (Sa:So) used as a biomarker. Fumonisins fed at 20.2 mg/kg for 14 days were responsible for a 4.4 fold increase in the Sa:So ratio and a decrease of 33% and 36% in C14-C16 ceramides and C14-C16 sphingomyelins, respectively, whereas C18-C26 ceramides and C18-C26 sphingomyelins remained unaffected or were increased. Glucosyl- and lactosyl-ceramides paralleled the concentrations of ceramides. Fumonisins also increased dihydroceramides but had no effect on deoxysphinganine. A partial least squfares discriminant analysis revealed that all changes in sphingolipids were important in explaining the effect of fumonisins. Because deoxynivalenol and zearalenone are often found in feed, their effects on sphingolipids alone and in combination with fumonisins were investigated. Feeding 5.12 mg deoxynivalenol/kg reduced dihydroceramides in the liver. Zearalenone fed at 0.47 mg/kg had no effect on sphingolipids. When fusariotoxins were fed simultaneously, the effects on sphingolipids were similar to those observed in turkeys fed fumonisins alone. The concentration of fumonisin B1 in the liver of turkeys fed fumonisins was 0.06 µmol/kg. Changes in sphingolipid concentrations differed but were consistent with the IC50 of fumonisin B1 measured in mammals; these changes could explain the relative resistance of turkeys to fumonisins.

## 1. Introduction

Fumonisins are mycotoxins produced by fungi of the genus *Fusarium*, and the worldwide contamination of food and feed by fumonisins is recognized as a serious problem for human and animal health [1,2,3]. Target organs and toxic effects of fumonisins vary markedly depending on the animal species and the dose and duration of exposure and fumonisins B are classified as a possible human carcinogen (2B) [4,5,6]. The molecular mechanisms of action of fumonisins are mainly attributed to disruptions of the sphingolipid metabolism, although the role of these alterations compared to other molecular changes is still the subject of debate [4,7,8,9,10]. However, the accumulation of sphinganine due to the inhibition of ceramide synthases (CerS) by the structural analogy of FB1 with the sphingoid bases is recognized as a key mechanism in fumonisin toxicity (Figure 1). An increase in free sphinganine in the liver and plasma occurs before toxicity, and the sphinganine to sphingosine ratio (Sa:So) is commonly used as a biomarker of fumonisins in animals and humans [4,5,11]. The alteration of the Sa:So ratio is not the only consequence of the inhibition of CerS, which also reduces ceramides, dihydroceramides, glycosyls ceramides, and sphingomyelins (Figure 1) [4]. Six CerS that vary in their carbon chain length specificity and in their expression in tissues have been characterized to date. CerS2, and to a lesser extent CerS4 and CerS5, are most abundant in the liver [12]. CerS2 is responsible for the synthesis of C20-C26 ceramides while CerS4 and CerS5, respectively, synthesize C18-C20, and C14-C16 ceramides [12]. The relative abundance of the different ceramides in cells has been shown to be important for the fate of the cell in apoptosis or proliferation [12,13]. Interestingly, it has been observed that changes in the concentration of sphingolipids and signs of fumonisin toxicity in mammals resembled those observed in CerS2 knockout mice [4,14,15]. Not only 1-deoxysphinganine and corresponding ceramides but also dihydroceramides could play a role in toxicity (Figure 1) [16,17]. Because sphingolipid metabolism in cells is complex, the inhibition of de novo synthesis alone is not sufficient to understand all the effects fumonisins have on sphingolipids. Activation of the sphingomyelin hydrolysis and recycling pathway (Figure 1) also explains the increase in the concentrations of ceramides and sphingosine reported in some studies [4] and could play a key role in fumonisin toxicity [18,19,20,21,22]. Therefore, the fine characterization of sphingolipid alteration is necessary to understand the multiple facets of fumonisin toxicity, including differences in sensitivity among species.

Among animal species, avian species are known to be relatively resistant to fumonisins [5,10]. Fumonisins in feed at a concentration of 20 mg/kg for several days to several weeks are toxic for pigs but not for turkeys, while variations in the Sa:So ratio in the liver and plasma are similar in the two species [23,24,25,26,27,28,29,30]. The relative resistance of avian species cannot be explained by differences in the oral bioavailability of fumonisins, which is close to that reported in mammals [31]. Moreover, recent studies have indicated that fumonisins persist for a long time in the liver of chickens and turkeys and that toxins cumulate in the body over time [32,33]. Increases in free sphinganine and its 1-phosphate derivate are similar in avian species and in mammals [34,35], and measuring sphingoid bases alone is not sufficient to explain the relative resistance of avian species to fumonisin toxicity. The development of quantitative LC-MS/MS analysis based on the high-efficiency chromatographic separation of analytes now enables the quantitation of low-abundance but biologically important sphingolipids [36]. When used in chickens fed fumonisins, this method has revealed numerous changes in the concentrations of sphingolipids in the liver [35]. Because turkeys are reported to be more sensitive to fumonisins than chickens [5,37], turkeys appear to be of interest to characterize the alterations of sphingolipids. What is more, the contamination of food and feed by mycotoxins is often multiple, and fumonisins, deoxynivalenol, and zearalenone are found together in feed destined for avian species, but little is known about the effect of interactions among fusariotoxins on sphingolipids [3,38]. Deoxynivalenol has been reported to reduce the de novo synthesis of sphingolipids and to modulate the recycling of sphingolipids due to its effect on the gut microbiome (Figure 1) [39,40,41]. Interactions between fumonisins, deoxynivalenol, and zearalenone have been reported in the membrane lipid profiles of rats and piglets [42,43]. The first purpose of this study was to characterize variations in the concentrations of sphingolipids in the liver of turkeys fed fumonisins to identify changes in sphingolipids and whether these changes explain the relative resistance of turkeys to fumonisins. Our second purpose was to investigate the effects of deoxynivalenol and zearalenone on sphingolipids and to characterize their interactions with fumonisins.

## 2. Results

The concentrations of sphingolipids in the liver of turkeys fed the mycotoxin-free control diet and the diets containing fusariotoxins are reported in Table 1 per analyte, and in Figure 2 as the sum of the different sphingolipids within a class. The sphingoid bases 1-phosphates, ceramide 1-phosphate, deoxysphingosine (dSo), and lactosylsphingosine (LacSo) were only found at trace level in this study.

Significant differences between the sphingolipid concentrations measured in turkeys fed the control diet and in turkeys fed the diets containing fusariotoxins were observed by ANOVA for many analytes, and these effects appeared to vary depending on the sphingolipid class but also the sphingoid base and the chain length of the fatty acid (Table 1). In contrast, the ANOVA conducted to compare the concentrations of the sphingolipid classes in the different groups revealed only significant differences between turkeys fed the control diet and turkeys fed diets containing fumonisins. In addition, no differences in ceramide levels were observed between turkeys fed the control diet and turkeys fed the diets containing fumonisins when a comparison was performed at the class level, whereas significant differences between groups were observed in 18:1/14:0, 18:1/16:0, 18:1/18:1, 18:1/23:0, 18:1/24:0, and 18:1/25:1 levels (Table 1). This difference is due to the fact that C14-C16 ceramides were decreased upon exposure to fumonisins while C18-C26 ceramides were unchanged or increased. Therefore, the presentation of the effects of fusariotoxins on sphingolipid levels in this study was done per mycotoxin and per sphingolipid.

### 2.1. Sphingolipids in the Liver of Turkeys Fed Fumonisins

The concentrations of sphingolipids in the liver of turkeys fed the control (CON) diet free of mycotoxins and the fumonisin diet (FB) that contained 20.2 mg FB1 + FB2/kg are reported in Table 1. Deoxysphinganine (dSa), glucosylsphingosine (GluSo), and lyso-sphingomyelin (LysoSM) were not affected by fumonisins. Together, sphinganine; N-acetyl-sphinganine (18:0/2:0); very-long-chain ceramides 18:1/23:0, 18:1/24:0 and 18:1/25:1; very-long-chain dihydroceramides 18:0/22:0, 18:0/23:0, 18:0/24:0; long- and very-long-chain sphingomyelins SM18:1/20:0, SM18:1/22:0, SM18:1/23:0, SM18:1/24:0, and SM18:1/25:0; and long- and very-long-chain dihydrosphingomyelins SM18:0/20:0, SM18:0/22:0, SM18:0/23:0, SM18:0/24:1, and SM18:0/24:0 increased significantly in the liver of turkeys fed fumonisins. By contrast, the effects of fumonisins on medium–long-chain sphingolipids varied with the sphingoid base. Together, ceramides 18:1/14:0 and 18:1/16:0, glycosylceramides glu18:1/16:0 and lac18:1/16:0, and sphingomyelin SM18:1/16:0, all containing sphingosine (d18:1), decreased significantly, whereas medium–long-chain dihydroceramide 18:0/16:0, and dihydrosphingomyelin SM18:0/16:0, which contain sphingosine (d18:0), increased significantly. The effects of fumonisins on sphingolipids whose fatty acid has one or more points of unsaturation generally paralleled those observed on sphingolipids whose fatty acid of the same chain length was saturated, but the difference between controls was not statistically significant.

The effects of fumonisins per class of sphingolipids and as a function of carbon chain length are summarized in Figure 3. A 4.4 fold increase in the sphinganine to sphingosine ratio (Sa:So) was observed in turkeys fed FB (Figure 3A), whereas no difference in the total concentration of sphingolipids in the liver was observed between groups (Figure 3B). Comparison of the relative abundance of the different classes of sphingolipids revealed that medium–long-chain C14-C16 ceramides decreased in turkeys fed FB, whereas very-long-chain C22-C28 sphingomyelins increased (Figure 3C). The concentration of C14-C16 ceramide was 33% lower in turkeys fed FB than in turkeys fed the CON diet. The C14-C16:C22-C26 ceramide ratio in the liver of turkeys fed the CON and the FB diets were 0.87 and 0.52, respectively. The effects of fumonisins on other sphingolipids were only slight or were too difficult to evaluate when measuring their relative abundance in the liver (Figure 3C).

A partial least squares discriminant analysis (PLS-DA) was conducted on sphingolipids dosed in the liver of turkeys fed the FB and the CON diets. Variables important in the projection (VIP), whose scores were above 1.1 for the two first components, are reported in Figure 4A,B, respectively. These variables generally corresponded to the sphingolipids for which a significant effect of fumonisins has already been reported (Table 1). Notably, together, sphinganine, ceramides, dihydroceramides, glycosylceramides, sphingomyelins, and dihydrosphingomyelins were important in the projection, showing that all the classes of sphingolipids in the liver were important in the analysis. These variables made it possible to clearly separate the turkeys into two different groups according to the diet they received (Figure 4C). The value of Q^2^ with the two first components was 0.916, indicating the very good quality of the model (Figure 4D). The values of the R^2^Y and the R^2^X indices were, respectively, 0.896 and 0.76, demonstrating that the selected sphingolipids very satisfactorily predicted the group to which the turkeys belonged, and were confirmed by the confusion matrix, which revealed that the model was highly sensitive and specific (Figure 4D).

### 2.2. Sphingolipids in the Liver of Turkeys Fed Deoxynivalenol and Zearalenone

The concentrations of sphingolipids in the liver of turkeys fed the deoxynivalenol diet (DON) that contained 5.12 mg DON/kg are reported in Table 1. Significant differences in the concentrations of sphingolipids in turkeys fed the CON and the DON diets were only observed for a few analytes. A significant increase in GluSo and a significant decrease in dihydroceramides 18:0/20:0 and 18:0/24:0 were observed in turkeys fed the DON diet. A numerically non-significant decrease in N-acetyl-sphinganine (18:0/2:0), and dihydroceramides 18:0/16:0, 18:0/18:0, 18:0/22:0, and 18:0/23:0 occurred in the liver of turkeys fed the DON diet, whereas the concentrations of ceramides, glycosylceramides, sphingomyelins, and dihydrosphingomyelins were close in the two groups.

The PLS-DA of sphingolipids dosed in the liver of turkeys fed the DON and the CON diets revealed that the four sphingolipids with the highest VIP score for the two first components corresponded to dihydroceramides (Figure 5A,B). Other sphingolipids that were important in the projection were So, Sa, GluSo, LysoSM, lac18:1/18:0, and ceramides. Interestingly, no sphingomyelin and no dihydrosphingomyelin had a VIP score above 1.1. As shown in Figure 5C, this analysis made it possible to clearly separate the turkeys into two groups. The values of Q^2^, R^2^Y, and R^2^X using the two first components, were, respectively, 0.779, 0.829, and 0.619, indicating the good quality of the model as confirmed by the results of the confusion matrix (Figure 5D).

Table 1 also shows the effects of zearalenone on the concentration of sphingolipids in the liver of turkeys when the toxin was fed at a concentration of 0.47 mg/kg (ZEN diet). Whatever the analyte dosed, no significant differences in the concentrations of sphingolipids were observed between turkeys fed the ZEN and the CON diet. Moreover, PLS-DA did not make it possible to predict whether the turkeys belonged to the CON or to the ZEN group, as revealed by the low value of the Q^2^cum obtained (Appendix A).

### 2.3. Sphingolipids in the Liver of Turkeys Fed A Combination of Fumonisins, Deoxynivalenol, and Zearalenone

Sphingolipids in the liver of turkeys fed fumonisins, deoxynivalenol, and zearalenone administered in a mixture at respective concentrations of 25.7, 5.15, and 0.57 mg/kg of feed (FDZ diet) are reported in Table 1. Almost all the differences in sphingolipid concentrations observed between turkeys fed the CON and the FB diet were again observed. The decrease in the concentration of dihydroceramides observed with the DON diet was not observed with the FDZ diet; instead, increases in d18:0/18:0, d18:0/20:0, d18:0/22:0, d18:0/23:0, and d18:0/24:0 were observed. This increase corresponded to the increase observed with the FB diet.

A PLS-DA was conducted on sphingolipids dosed in the liver of turkeys fed the FB, DON, ZEN, FDZ, and CON diets according to the presence (yes) or absence (no) of fumonisins in the diets (Figure 6). Most of the sphingolipids whose VIP score was above 1.1 on the two first components (in red in Figure 6A,B) corresponded to sphingolipids that were important in the projection when the PLS-DA was conducted on the FB and the CON diets (Figure 4). Only two sphingolipids with a VIP score above 1.1 (in blue in Figure 6A,B) corresponded to compounds that were important when the PLS-DA was conducted on the DON and the CON (Figure 5). Seven sphingolipids were important in both analyses (in purple in Figure 6A,B). This analysis enabled a very good separation of the turkeys into two groups (Figure 6C). The values of Q^2^, R^2^Y, and R^2^X were 0.908, 0.91, and 0.753, respectively. These values were very high, indicating the very good quality of the model, which was confirmed by the results of the confusion matrix (Figure 6D).

Because the FDZ diet also contained DON, another PLS-DA was conducted according to the presence (yes) or absence (no) of deoxynivalenol in the diets (Appendix A). The VIP in this analysis mainly corresponded to VIPs whose scores were above 1.1 when a PLS-DA was conducted on the DON and CON diets. However, only a partial separation of turkeys was obtained (Appendix A). Moreover, the values of Q^2^, R^2^Y, and R^2^X in this analysis were 0.302, 0.48, and 0.534, respectively, pointing to the lesser quality of the model (Appendix A). Finally, this analysis confirmed that the impact of deoxynivalenol on sphingolipids was weak when fumonisins and deoxynivalenol were present together in the diet.

A final PLS-DA was conducted according to the five experimental diets (Appendix A). VIP scores that were above 1.1 in this analysis were the same as those found when the PLS-DA was conducted according to the presence or absence of fumonisins in the diets (Figure 5). A good value of X^2^ of 0.754 was obtained, whereas the values of R^2^Y and Q^2^ were 0.232 and 0.172, respectively (Appendix A). These low values showed that the sphingolipids (X) did not explain the distribution of the turkeys in the different groups (Y), thereby confirming the very strong effect of fumonisins on sphingolipids compared to other fusariotoxins.

### 2.4. Correlations among Sphingolipids in the Liver of Turkeys Fed Fusariotoxins

Correlations among sphingolipid concentrations in the liver were investigated in the five groups corresponding to 48 turkeys, 20 of which were fed fumonisins alone or mixed with deoxynivalenol and zearalenone, and 28 received a diet that contained no fumonisins. A significant positive correlation was found between the concentrations of C14 and C16 ceramides and between the concentrations of C20 to C24 ceramides. By contrast, no correlation was found between C14-C16 ceramides and C20-C24 ceramides. The correlations between C18 ceramides and other ceramides were significant but lesser than the correlations between C14-C16 and C20-C24 (Table 2).

The concentrations of all the dihydroceramides dosed were positively correlated, whatever the chain length of the fatty acid involved in their composition (Table 3). The concentrations of ceramides whose fatty acid was unsaturated were generally correlated with the concentrations of ceramides containing a saturated fatty acid of the same chain length (Appendix A).

### 2.5. Fumonisins in the Liver of Turkeys Fed the Different Diets

The mean concentrations of FB1, FB2 and FB3 in the liver of turkeys fed the FB diet and in turkeys fed the FDZ diet were 60.58, 5.81, 2.29, and 60.89, 5.59, 2.56 nmol/kg, respectively (Table 4). FB1, FB2, and FB3 were below the LOQ of 0.35 nmol/kg in the liver of turkeys fed the control diet free of mycotoxins and the diets containing zearalenone and deoxynivalenol.

## 3. Discussion

The effects on the health of the different diets fed in this study are reported in [23]. No effects of fusariotoxins alone or in combination were observed on several variables used to reveal toxicity, consistent with the regulatory guidelines on the maximum tolerable level of fusariotoxins in avian feed [5,44,45,46]. The measurement of the effect of fusariotoxins on sphingolipid levels was carried out in the liver, which is a target organ of fumonisins and for which the largest number of results is available to allow a comparison of effects between studies and between species.

### 3.1. Effects of Fumonisins

#### 3.1.1. Sphingoid Bases

The marked increase in the Sa:So ratio in the liver of turkeys fed 20.2 mg of FB1 + FB2/kg fumonisins in the absence of toxicity agrees with previous results obtained in this animal species [23,24,25,26,27,47,48,49]. Deoxysphinganine was not affected by fumonisins in the present study, in agreement with results obtained in chickens fed a 20.8 mg of FB1 + FB2/kg diet for 9 days [35]. Deoxysphinganine has been found to be increased by FB1 in mammals and is recognized as a key toxic metabolite of sphinganine that, along with deoxyceramides, could contribute to fumonisin toxicity [4,17,50]. The lack of an effect of fumonisins on deoxysphinganine concentrations could be a first explanation for the relative resistance of turkeys to fumonisins.

#### 3.1.2. Dihydroceramides and Dihydrosphingomyelins

All dihydroceramides, including N-acetylsphinganine (18:0/2:0), and all dihydrosphingomyelins were increased in turkeys fed fumonisins, most of these increases being significant. This result is in agreement with results reported in chickens fed 20.8 mg of FB1 + FB2/kg for nine days [35]. Because fumonisins are inhibitors of (dihydro)ceramide synthases, a decrease was expected [4]. However, it should be noted that not all the studies reported decreased concentrations of dihydrosphingolipids [28,51]. A convenient mechanism to explain the increase in dihydrosphingolipids would be that fumonisins inhibit dihydroceramide desaturase (Figure 1). The inhibition of dihydroceramide desaturase by xenobiotics such as synthetic vitamin A analog, fenretinide, ST1926, and nonflavonoid polyphenol resveratrol has been reported [52,53,54]. It has also been reported that fumonisins inhibit other desaturases such as delta6-desaturases [20,55,56]. Another mechanism that could explain the increased concentrations of dihydroceramides in the liver is related to the different affinity of (dihydro)ceramide and ceramide synthases for sphinganine, sphingosine, and acyl-CoA [57]. As a 4.4 fold increase in sphinganine occurred in the present study, whereas the concentration of sphingosine was not altered by fumonisins, it can be hypothesized that larger amounts of acyl-CoA were used to form dihydroceramides by the de novo synthesis of sphingolipids than ceramides via salvage pathways (Figure 1).

Whatever the mechanism, the increase in dihydroceramides in the liver of turkeys fed fumonisins could have consequences for health. Indeed, dihydroceramides are a subclass of sphingolipids that were long considered to have only weak biological activity, whereas recent studies highlight their roles in diseases and cell survival [16]. Notably, inhibitors of dihydroceramide desaturase prevent or reverse the pathogenic features of nonalcoholic fatty liver steatosis and cardiomyopathy, signs of toxicity reported in some studies on fumonisins [37,58,59,60]. Moreover, the resaturation of dihydroceramide concentrations in cells was recently correlated with the reduced toxicity of different toxins [61].

#### 3.1.3. Ceramides

A decrease in the total concentration of ceramides in the liver was observed in the present study; the decrease corresponded to a decrease in C14-C16 ceramides, whereas C18-C26 ceramides were either not affected or were increased. These results are in agreement with those obtained in chickens fed 20.8 mg of FB1 + FB2/kg for nine days [35]. The 33% decrease in C14-C16 ceramides measured in this study was observed at a concentration of 0.06 µmol FB1/kg in the liver, which is consistent with the IC50 of 0.1 µmol FB1/L measured in the ceramide synthases (CerS) in rat hepatocytes in culture [62].

Even if little is known about CerS in avian species, recent studies on Brown Tsaiya ducks suggest that the enzymes involved in the synthesis of sphingolipids were the same as those reported in mammals [63]. CerS2, and to a lesser extent CerS4 and CerS5, are most abundant in the liver and are responsible for the synthesis of, respectively, C20-C26, C18-C20, and C14-C16 ceramides (Figure 1) [12]. The fact C14-C16 ceramides were decreased in turkeys fed fumonisins, whereas C18-C26 ceramides were not affected or were increased, suggests the selective inhibition of CerS5. By contrast, the increase in C18-C26 ceramides could be a compensatory mechanism, as already demonstrated in knockout mice [14]. The hypothesis that C14-C16 ceramides and C18-C26 ceramides are differentially affected and regulated in the presence of fumonisins in turkeys is reinforced by the correlations observed between these analytes (Table 2). The very strong positive correlations observed between C14-C16 ceramides and C20-C25 ceramides—whereas C14-C16 and C20-C25 ceramides were not correlated—and the intermediate correlations observed between C18 ceramides and other ceramides are consistent with what is already known about the specificity of CerS5, CerS2, and CerS4 in mammals (Figure 1).

The effects of fumonisins on ceramides could be a key to explaining toxicity. Indeed, the overexpression of C16 ceramides has been reported to have pro-apoptotic effects in irradiated HeLa cells, whereas the overexpression of C24 ceramides was anti-apoptotic [64]. Very-long-chain ceramides also reduced the permeabilization of rat liver mitochondria mediated by C16-ceramides [65]. Further, it has been suggested that the changes in sphingolipids observed in CerS2 knockout mice (mainly an increase in C16 ceramides and a decrease in C22-C24 ceramides) resembled the effects of FB1 in mammals, including toxicity for the liver [15]. By contrast, the effects of fumonisins on sphingolipids in this study on turkeys, including the effects on dihydroceramides, glycosylceramides, and sphingomyelins, resemble the changes to sphingolipids observed in CerS5 knockout MCF-7 human breast adenocarcinoma cells [66].

#### 3.1.4. Glycosylceramides

The concentrations of glucosylceramides, hexosylceramides, and lactosylceramides in this study generally paralleled the concentrations of the corresponding ceramides in agreement with results obtained in chickens [35]. This result is consistent with the synthesis pathways of glycosylsphingolipids (Figure 1). In mouse cerebellar neurons in culture, the IC50 of glycolipid synthesis was shown to be approximately 7 µmol of FB1/L, which is considerably higher than the concentration of 0.06 µmol of FB1/kg found in the liver of turkeys [67].

#### 3.1.5. Sphingomyelins

The concentrations of sphingomyelins in the liver of turkeys were generally proportional to the concentrations of the corresponding ceramides: C16 sphingomyelins decreased, whereas C20-C25 sphingomyelins increased. Because very-long-chain sphingomyelins were the most abundant in the liver of turkeys, the total concentration of sphingomyelins was increased. These results are in agreement with those obtained in chickens fed 20.8 mg of FB1 + FB2/kg for nine days [35]. The lack of a decrease in sphingomyelins at a concentration of 0.06 µmol of FB1/kg in the liver in this study is consistent with the IC50 of 0.7 µmol of FB1/L measured in mouse cerebellar neurons in culture for the synthesis of sphingomyelins [67]. A reduction in sphingomyelins in cells can also be the result of the activation of sphingomyelinases (Figure 1). The activation of acid sphingomyelinase has been obtained in mice by the subcutaneous administration of FB1 at hepatotoxic doses [18]. Sphingomyelins tended to decrease in the liver of pigs administered FB1 orally at a dose equivalent to 25–30 mg/kg of feed, and moderate signs of toxicity were observed at this dose [28]. A reduction in sphingomyelin concentrations at cytotoxic doses of FB1 was also observed after the direct administration of FB1 to Brown Tsaiya duck embryos [63], and in cell cultures at concentrations of 20 to 250 µM of FB1 [20,21,22]. Moreover, the role of sphingomyelin in FB1-induced toxicity has been observed in mice using myriocin, which is a specific inhibitor of serine palmitoyltransferase (Figure 1). Myriocin completely prevented any increase in free sphinganine due to FB1, in agreement with previous in-vitro studies [68], but did not prevent a decrease in sphingomyelin and toxicity in mice [51]. The lack of activation of the sphingolipid recycling pathway in the present study probably contributes to the absence of fumonisin toxicity in turkeys. This hypothesis is consistent with recent studies that demonstrated a decrease in sphingomyelins preceding ceramide-induced apoptosis [19].

### 3.2. Effects of Deoxynivalenol and Zearalenone Alone and Combined with Fumonisins

The consequences of feeding deoxynivalenol and zearalenone on sphingolipids in the liver of turkeys are difficult to compare, as only a few authors have investigated these effects. The lack of an effect of deoxynivalenol on the sphingoid bases in turkey agrees with the results of a study on broilers fed 1.5 mg/kg of feed that had no effect on the Sa:So ratio [69]. The reduction in dihydroceramide concentrations observed in the present study is consistent with a study conducted on yeast culture that revealed that acetylated deoxynivalenol down-regulated sphingolipid metabolism pathway genes, suggesting a decrease in the de novo synthesis of sphingolipids (Figure 1) [39]. However, it is difficult to draw concrete conclusions as both the model and the acetylated form of the toxins differed. Different studies on mice have shown that deoxynivalenol can modulate the relative abundance of *Bacteroidetes* in digesta [70,71]. Because the phylum *Bacteroidetes* is known to produce sphingolipids that can be recycled in the host (Figure 1), it was hypothesized that this mechanism could modulate sphingolipid concentrations in the liver [40,41,72]. However, the targeted sphingolipidomic analysis conducted in the present study did not reveal an effect of DON on sphingolipid recycling pathways. The lack of an effect of zearalenone on the sphingoid bases in this study is in agreement with results obtained using precision-cut slices of rat liver, which revealed that high doses of zearalenone had no effect on the Sa:So ratio [73].

The absence of an effect of interactions among fumonisins, deoxynivalenol, and zearalenone on the sphingoid bases in turkeys is in agreement with the results of studies of the Sa:So ratio in broilers and in ducks [74,75]. A study on broilers fed deoxynivalenol and fumonisins at doses of 1.5 and 20 mg/kg, respectively, also revealed no interaction on the Sa:So ratio [69]. Finally, all the effects of the mixtures of fusariotoxins observed on sphingolipids in this study corresponded to the effects of the fumonisins.

## 4. Materials and Methods

### 4.1. Analytes and Reagents

All the analytes and reagents used in this study were purchased from Sharlab (Sharlab S.L., Sentmenat, Spain) or Sigma (Sigma Chemical Co, Saint Quentin Fallavier, France). All the reagents were HPLC analytical grade except the pure water, methanol, isopropanol, and formic acid used for the mass analysis, which were LC-MS grade. The 33 sphingolipids used as standards were deoxysphingosine (dSo); deoxysphinganine (dSa); sphingosine (So = 18:1); sphinganine (Sa = 18:0); sphingosine-1-P (18:1P); sphinganine-1-P (17:1P); glucosylsphingosine (GluSo); lyso-sphingomyelin (LysoSM); lactosylsphingosine (LacSo); N-acetyl-sphingosine (18:1/2:0); N-acetyl-sphinganine (18:0/2:0); ceramides 18:1/14:0, 18:1/16:0, 18:1/18:0, 18:1/20:0, 18:1/22:0, 18:1/24:1, and 18:1/24:0; ceramide-1P (18:1/16:0P); dihydroceramides 18:0/16:0, and 18:0/24:0; glucosylceramides 18:1/16:0 (Glu18:1/16:0) and Glu18:1/24:1; lactosylceramides 18:1/16:0 (Lac18:1/16:0) and Lac18:1/24:1; and sphingomyelins (SM) SM18:1/14:0, SM18:1/16:0, SM18:1/18:0, SM18:1/18:1, SM18:1/20:0, SM18:1/22:0, SM18:1/24:1, and SM18:1/24:0. These were purchased in solid form from Sigma (Sigma Chemical Co, Saint Quentin Fallavier, France). Ten sphingolipids used as internal standards (IS) were purchased from Sigma in the form of “Ceramide/Sphingoid Internal Standard Mixture I” of Avanti Polar Lipids that contain 25 µM of sphingosine (C17 base), sphinganine (C17 base), sphingosine-1-P (C17 base), sphinganine-1-P (C17 base), lactosyl (ß) C12 ceramide, 12:0 sphingomyelin, glucosyl (ß) C12 ceramide, 12:0 ceramide, 12:0 ceramide-1-P, and 25:0 ceramide in solution in ethanol. Standard fumonisin B1, fumonisin B2, fumonisin B3, deoxynivalenol, desoxynivalenol-3-glucoside, deepoxy-deoxynivalenol, 15-acetyl-deoxynivalenol, 3-acetyl-deoxynivalenol, nivalenol, zearalenone, alpha-zearalenol, beta-zearalenol, zearalanone, alpha-zearalanol, beta-zearalanol, diacetoxyscirpenol, 15 monoacetoxyscirpenol, T2 toxin, HT2 toxin, T2 tetraol, verrucarol, verrucarin A, fusarenone x, roridin A, moniliformin, tenuazonic acid, ergocornine, ergocristine, ergocryptine, ergometrine, ergosine, ergotamine, aflatoxin B1, aflatoxin B2, aflatoxin G1, aflatoxin G2, sterigmatocystin, ochratoxin A, ochratoxin B, alpha-ochratoxin, cyclopiazonic acid, citrinin, and patulin were purchased from Biopure (Romer Labs, 3131 Getzersdorf, Austria) or Sigma. Standard certified solutions of [^13^C_34_]-FB1, [^13^C_34_]-FB2 were purchased from Biopure. FUMONIPREP columns were purchased from R-Biopharm (R-Biopharm Rhone LTD, Glasgow, Scotland).

### 4.2. Animals and Treatment

All experimental procedures with animals were conducted in accordance with the French National Guidelines for the care and use of animals for research purposes and the guidelines of the Declaration of Helsinki using an experimental protocol approved by the French Ministry of Higher Education and Research, registered under number 02032.01. The biological samples used in this study were obtained in a study in turkeys for which the feed formulation, experimental animal protocol, and the effects of mycotoxins on health are detailed in [23].

Briefly, fumonisins, deoxynivalenol, and zearalenone were obtained by culturing toxigenic strains of *Fusarium verticillioides* on corn and two strains of *F. graminearum,* strain I159 and strain I171, on wheat and on rice, respectively. At the end of the culture period, the cultured material was dried and concentrations of mycotoxins were measured by LC-MSMS according to AFNOR V03-110. Culture materials were added to the feed made of a mixture of corn and soybean formulated to best meet turkeys’ nutritional needs. Five experimental diets were formulated: a control diet (CON) free of mycotoxins, a fumonisin diet (FB) with a target concentration of 20 mg of FB1 + FB2/kg of feed, a deoxynivalenol diet (DON) with a target concentration of 5 mg/kg of feed, a zearalenone diet (ZEN) with a target concentration of 0.5 mg/kg of feed, and a diet that contained fumonisins, deoxynivalenol, and zearalenone (FDZ) with respective concentrations of 20, 5, and 0.5 mg/kg of feed. The final concentrations of fusariotoxins (Table 1) and the absence of other mycotoxins that could interfere with the results were measured in the diet before the assays with animals.

Male Grade Maker turkeys reared in individual cages were fed and watered ad libitum at the ARVALIS-*Institut du végétal* experimental research station (41000 Villerable, France). The turkeys were fed with mycotoxin-free diets from 0 to 55 days of age. On the 55th day of age, the turkeys were weighed and randomly assigned to one of five groups to be fed with the five different experimental diets. On the 70th day of age, and after a starvation period of 8 h, the turkeys were killed by exsanguination after stunning by electrocution. All the animals were autopsied and their livers were removed and stored at −80 °C until analysis. No effect of feeding fusariotoxins was observed in analyses of performance, organ weight, biochemistry, histopathology, oxidative damage, or testis toxicity [23].

### 4.3. Chromatographic System

The UPLC MS/MS system used for the analysis of sphingolipids and fumonisins in the liver was composed of a 1260 binary pump, an autosampler, and an *Agilent 6410 Triple Quadrupole* Spectrometer (Agilent, Santa Clara, CA, USA). Analytes were separated using a Poroshell 120 column (3.0 × 50 mm, 2.7 µm). Detection was performed after positive electrospray ionization at 300 °C at a flow rate of 10 L/min under 25 psi and 4000 V capillary voltage. Agilent MassHunter Optimizer software was used for the optimization of the transitions, fragmentor voltages, and collision energies of the analytes available as standards. The parameters obtained for the sphingolipids available as standards were also used for sphingolipids of the same class that were not available as standards. The chromatograms were analyzed using Agilent MassHunter quantitative analysis software. Accuracy was considered acceptable for a relative standard deviation (RSD) of 20%. Signal suppression and enhancement (SSE) were calculated using the slope method for fumonisins and the area method for sphingolipids. Values of SSE outside 80–120% indicated that a matrix effect had occurred, which was taken into account in the calculation of the final concentration of the analytes in the liver. Intra-day and inter-day repeatability were evaluated by the RSD of the recovery measured on the IS, which should be below 20%. Samples with analyte concentrations 25% above the maximum limit validated were diluted prior to re-analysis. If the acceptability criteria were not met, the results were discarded and samples were re-analyzed.

### 4.4. Sphingolipid Analysis

The method of the analysis of sphingolipids is described in [35]. The analytes were separated on the UHPLC-system with a gradient of elution composed of methanol/acetonitrile/isopropanol (4/1/1, *v*/*v*/*v*) and pure water, each containing 10 mM ammonium acetate and 0.2% formic acid (*v*/*v*) delivered at a flow rate of 0.3 mL/min. MRM parameters and retention times are detailed in [35]. Briefly, the m/z of the product ions used for quantification were 1dSo 266.4; d17:1 268.4; 1dSa 268.3; d17:0 270.4; d18:1 282.3; d18:0 284.3; 18:1/2:0 264.3; 18:0/2:0 266.3; d17:1P 250.3; d17:0P 270.3; d18:1P 264.3; d18:0P 284.3; GluSo 282.3; LysoSM 184; LacSo 282.3; 18:1/12:0 and all ceramides 264.3; 18:0/16:0, 18:0/18:0 and 18:0/20:0 284.3; 18:0/22:0, 18:0/23:0, and 18:0/24:0 266.3; Glu18:1/12:0, Lac18:1/12:0 and all glycosylceramides 264.3; SM18:1/12:0 and all sphingomyelins and all dihydrosphingomyelins 184. Retention times ranged from 9 min for d17:1 to 24.5 min for 18:1/26:0.

The liver samples were prepared as described in [35]. Briefly, 1 g of liver was homogenized with a potter in 3 mL phosphate buffer (0.1 M, pH 7.4). After 15 min centrifugation at 3000× *g*, 40 µL of the supernatant was collected and 120 µL of 0.9% NaCl, 600 µL of methanol/chloroform (2/1, *v*/*v*), and 10 µL of a solution containing the 10 IS in ethanol at a concentration equivalent to 6250 pmol/g of liver were added. Samples were incubated at 48 °C overnight. Alkaline hydrolysis was conducted to cleave glycerophospholipids by adding 75 µL of methanol KOH (1M). After 2 h of incubation at 37 °C, KOH was neutralized with 8 µL of 50% acetic acid and the samples were centrifuged at 7000× *g* for 15 min. The supernatant was collected and the residue was extracted again. The two supernatants were pooled and evaporated to dryness. The dry residue was solubilized in 200 µL of MetOH and 5 µL were injected into the chromatographic system.

The linearity of the method of analysis for the sphingolipids available as standards was good over a relatively large range of concentrations (Appendix A), in agreement with previous results [35,36]. Intra-day and inter-day repeatability were satisfactory, as attested by the RSD measured on the IS that was generally below 20% (Appendix A). The mean RSD measured for ceramide 18:1/25:0 was slightly above 20%, in agreement with previous results, and was considered acceptable [35]. The concentrations of sphingolipids available as standards were calculated by quadratic calibration using 1/x^2^ weighting factor, while the concentrations of sphingolipids not available as standards were calculated using the calibration curves obtained for standards in the same class with the closest mass. The final concentrations of sphingolipids in the liver were corrected by the recovery measured for the IS as follows: d17:1 for dSo and d18:1, d17:0 for dSa and d18:0; d17:1P for d18:1P; d17:0P for d18:0P; 18:1/12:0 for 18:1/14:0, 18:1/16:0, 18:0/16:0, 18:1/18:1, and 18:1/18:0; 18:1/25:0 for all ceramides and all dihydroceramides whose fatty acid chain length was C20 and above; SM18:1/12:0 for all sphingomyelins and all dihydrosphingomyelins; Glu18:1/12:0 for all glucosyl- and all hexosylceramides; Lac18:1/12:0 for all lactosylceramides. No correction by IS was used for 18:1/2:0, 18:0/2:0, GluSo, LysoSM, or LacSo.

### 4.5. Measurement of Fumonisins in the Liver

The analytical method used for the quantitation of FB1, FB2, and FB3 in the liver has already been validated with a limit of quantitation (LOQ) defined as the lowest concentration level validated of 0.25 ng/g equivalent to 0.35 nmol/kg for FB1, FB2, FB3 [76]. Briefly, one g of liver was homogenized with an Ultra Turrax in 2 mL of distilled water, 2 mL of acetonitrile/methanol (1:1), and 5 mg of NaCl and placed for 2 h on a stir table. After 15 min of centrifugation at 3000× *g*, the supernatant was collected and extracted with 8 mL of hexane. A portion of the aqueous phase was diluted in PBS and [^13^C_34_]-FB1 and [^13^C_34_]-FB2 were added to reach a final concentration equivalent to 12.5 ng/g of liver. Fumonisins were extracted using a FUMONIPREP column following the manufacturer’s instructions. The dry residue was suspended in 200 µL of the mobile phase and 10 µL were injected into the UPLC MS/MS system. The mobile phase was composed of a mixture of methanol and water, each containing 0.1% formic acid (*v*/*v*), and delivered using a gradient of elution as previously described [76]. MRM parameters and retention times are detailed in [76]. Briefly, the most abundant product ion of each precursor ion was used as a quantifier, two different ions were used as qualifiers for FB1 (m/z: precursor 722.4, quantifier 334.4, qualifier 352.4 and 704.4) and FB2 and FB3 (m/z: precursor 706, quantifier 336.4, qualifier 318.4 and 754.4). One qualifier was used for the [^13^C_34_]-FB1 (m/z: precursor 756.4, quantifier 356.5, qualifier 738.6) and [^13^C_34_]-FB2 (*m*/*z*: precursor 740, quantifier 358.6, qualifier 340.5). The qualifier ratio in the samples had to be less than 20% of the qualifier ratio measured in the standards. The method of analysis was linear, from 0.25 to 100 µg/kg for FB1 and from 0.25 to 5 µg/kg for FB2 and FB3 [76]. The concentrations of FB1, FB2, and FB3 were calculated by linear regression using the calibration curves obtained for the standards. The final concentrations in the liver were corrected by the recovery measured for the IS. The good repeatability of the method was attested by the RSD of the recovery measured on the IS, which was below 20%.

### 4.6. Statistical Analysis

All statistical analyses were performed using XLSTAT Biomed (Addinsoft, 33,000 Bordeaux, France). Sphingolipids and fumonisins in the liver are reported as means ± SD. Differences among groups were analyzed using one-way ANOVA after checking the homogeneity of variance (Hartley’s test). When a significant difference was observed among groups (ANOVA, *p* < 0.05), an individual comparison of means was conducted. Different letters in the same row indicate statistically different means (Duncan, *p* < 0.05). Partial least squares discriminant analysis (PLS-DA) was performed on the concentrations of sphingolipids measured in the liver according to the different diets fed to the five groups of turkeys. A score above 1.1 was used to select the variables important in the projection. Pearson’s correlation was used to measure the correlations between the sphingolipids dosed (*p* < 0.05).

## 5. Conclusions

In conclusion, the increase in the Sa:So ratio, the decrease in the total concentration of ceramides, and the effect of fumonisins on the total concentration of glycosylceramides and sphingomyelins observed in this study are consistent with the level of FB1 in the liver and its known IC50 measured in mammals on the *de novo* synthesis of sphingolipids. The effects of fumonisins on the different classes of ceramides were characterized by a decrease in C14-C16 ceramides, whereas C20-C26 ceramides were not affected or increased. Because of the role of C14-C16 and C20-C26 ceramides in the fate of the cells in apoptosis or proliferation, this could be a key point in explaining the lack of toxicity of this dose of fumonisins in turkeys. The lack of an increase in the concentration of deoxysphinganine and the increase in the concentrations of dihydroceramides could also contribute to the relative resistance of turkeys to fumonisin toxicity. SM18:1/16:0 was the only sphingomyelin to decrease and this decrease probably corresponded to a decrease in its synthesis resulting from a decrease in C16 ceramide. By contrast, C18-C26 sphingomyelins were not affected or increased. Because most of the studies conducted in mammals reported a decrease in sphingomyelins in the course of fumonisin toxicity resulting from the activation of salvage pathways, the absence of the activation of this pathway in turkeys in the present study probably contributed to the lack of fumonisin toxicity. Feeding deoxynivalenol was responsible for a small but significant decrease in the concentration of dihydroceramides in the liver. In this study, zearalenone had no effect on sphingolipids. When fusariotoxins were fed simultaneously, the effects of fusariotoxins on sphingolipids were similar to those observed in turkeys fed fumonisins alone, and these effects masked those observed with deoxynivalenol.

## Figures and Tables

**Figure 1 ijms-23-02512-f001:**
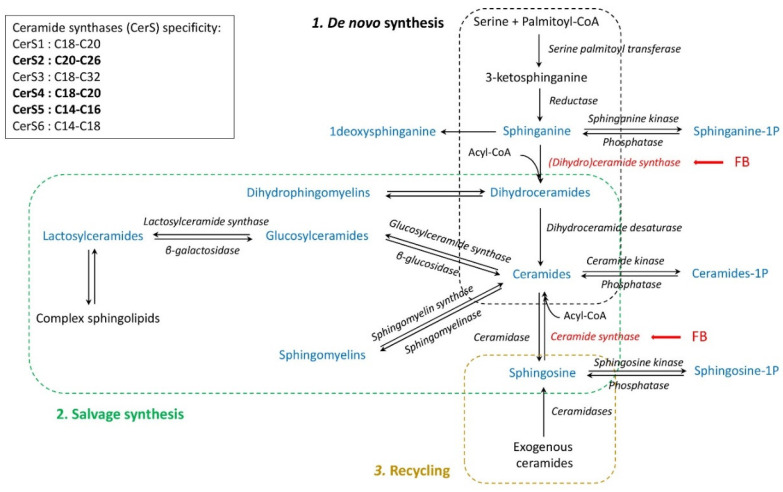
Different sphingolipid synthesis pathways. Fumonisin B (FB) inhibits ceramide synthases (red arrow) leading to marked changes in the sphingolipid profile. Different ceramide synthases have been identified in mammals, their specificity varied depending on the length of the carbon chain of the fatty acid. CerS2, CerS4, and CerS5 are the most abundant in the liver [4,12].

**Figure 2 ijms-23-02512-f002:**
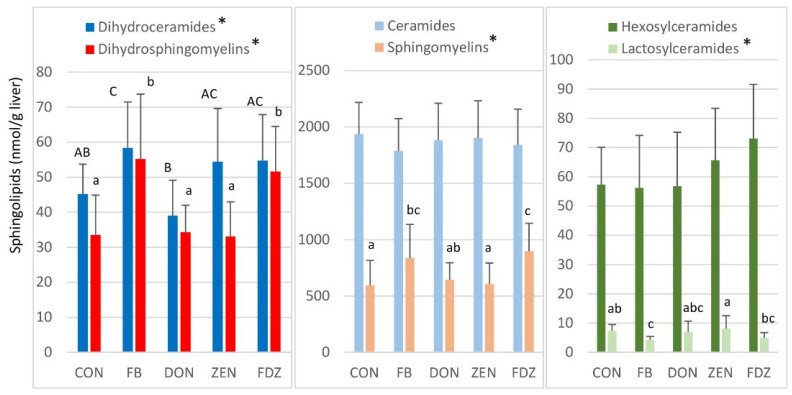
Concentrations of sphingolipids reported as the total of the different sphingolipids measured within a class in the liver of turkeys fed the mycotoxin-free control diet (CON) and different diets containing fumonisins (FB) at 20.2 mg/kg expressed as the sum of FB1 + FB2, deoxynivalenol (DON) at 5.12 mg/kg, zearalenone (ZEN) at 0.47 mg/kg, and fumonisins, deoxynivalenol, and zearalenone (FDZ). * Significant difference among groups (ANOVA, *p*< 0.05). Different letters in the same row indicate statistically different means (Duncan, *p* < 0.05).

**Figure 3 ijms-23-02512-f003:**
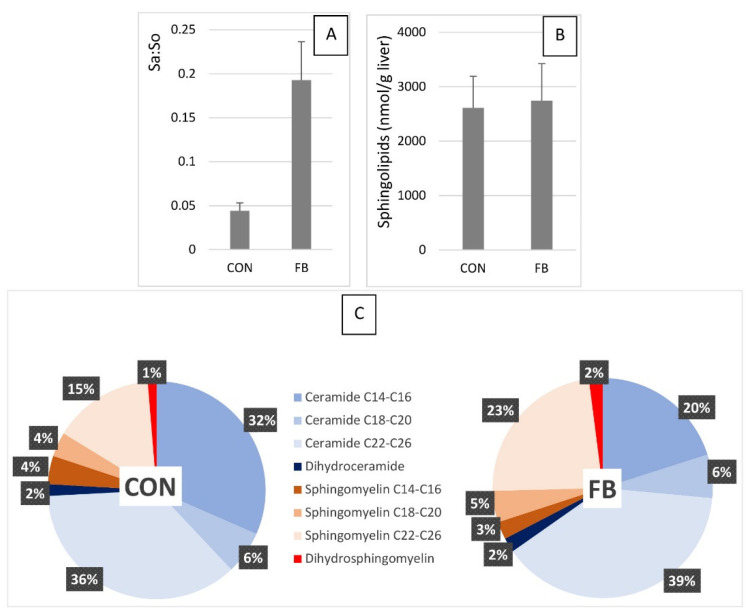
Sa:So ratio (**A**) and the total (**B**) and relative (**C**) abundance of the different classes of sphingolipids measured in the liver of turkeys fed a mycotoxin-free control diet (CON) or a diet containing fumonisins (FB) at a concentration of 20.2 mg FB1 + FB2/kg.

**Figure 4 ijms-23-02512-f004:**
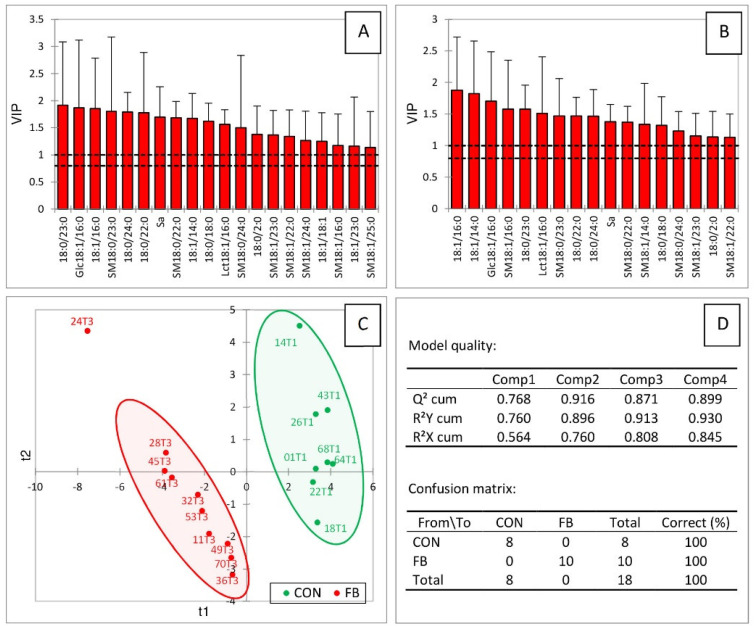
Partial least squares discriminant analysis (PLS-DA) of sphingolipids measured in the livers of turkeys fed a mycotoxin-free control diet (CON, T1) or a diet containing fumonisins (FB, T3) at a concentration of 20.2 mg FB1 + FB2/kg. Scores of the variables that are important in the projection (VIP) for the first (**A**) and the second (**B**) components. (**C**) Discrimination on the factor axes extracted from the original explanatory variables. (**D**) Quality of the model and confusion matrix for the training sample (variable groups).

**Figure 5 ijms-23-02512-f005:**
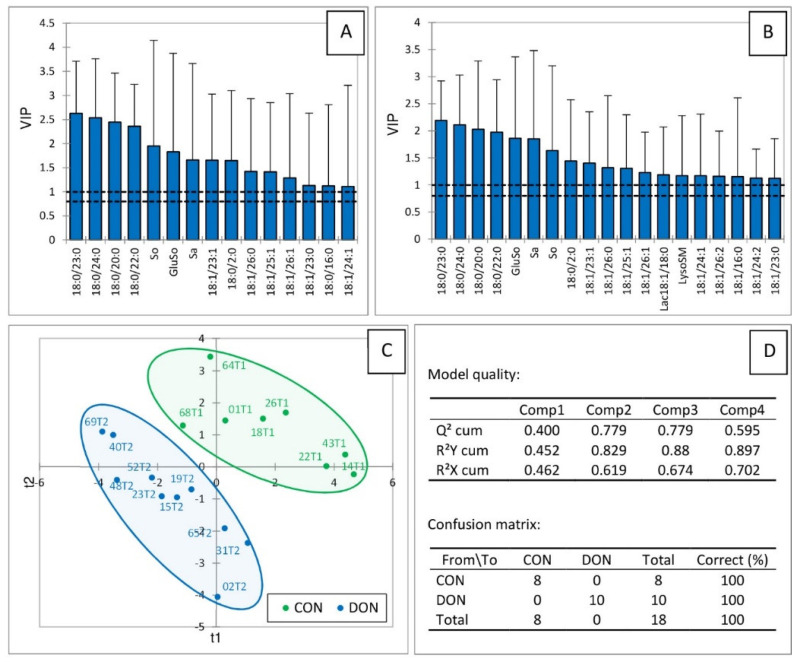
Partial least squares discriminant analysis (PLS-DA) of sphingolipids measured in the livers of turkeys fed a mycotoxin-free control diet (CON, T1) or a diet containing deoxynivalenol (DON, T2) at a concentration of 5.12 mg/kg. Scores of the variables that are important in the projection (VIP) for the first (**A**) and the second (**B**) components. (**C**) Discrimination on the factor axes extracted from the original explanatory variables. (**D**) Quality of the model and confusion matrix for the training sample (variable groups).

**Figure 6 ijms-23-02512-f006:**
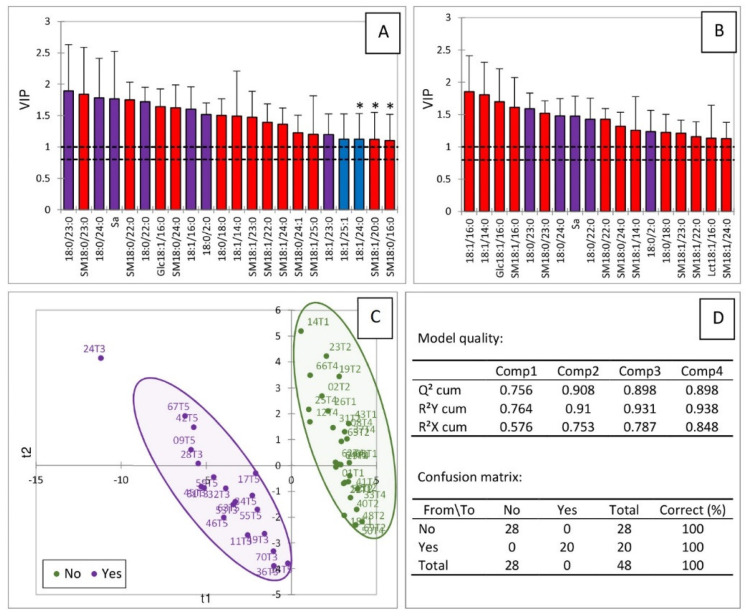
Partial least squares discriminant analysis (PLS-DA) of sphingolipids measured in the livers of turkeys fed 5 experimental diets according to the presence (yes) or absence (no) of fumonisins in their diet. The 5 experimental diets corresponded to a mycotoxin-free control diet (CON, T1), a diet containing fumonisins (FB, T3) at a concentration of 20.2 mg FB1 + FB2/kg, a diet containing deoxynivalenol (DON, T2) at a concentration of 5.12 mg/kg, a diet containing zearalenone (ZEN, T4) at a concentration of 0.47 mg/kg, and a diet containing fumonisins, deoxynivalenol, and zearalenone in combination (FDZ, T5) at respective concentrations of 25.7, 5.15, and 0.57 mg/kg of feed. Scores of the variables that are important in the projection (VIP) for the first (**A**) and the second (**B**) components. (**C**) Discrimination on the factor axes extracted from the original explanatory variables. (**D**) Quality of the model and confusion matrix for the training sample (variable groups). * 18:1/24:0 has a VIP score of 0.99 for the second component in the PLS-DA of sphingolipids obtained from chickens fed the DON and the CON diets. SM18:0/16:0 and SM18:1/20:0 have VIP scores of 1.037 and 1.068, respectively, for the first component in the PLS-DA of sphingolipids obtained from chickens fed the FB and the CON diets.

**Table 1 ijms-23-02512-t001:** Concentrations of sphingolipids in the liver of turkeys fed diets containing fusariotoxins ^1^.

	CON	FB	DON	ZEN	FDZ
Sphingoid bases and derivates ^2^					
dSa	24 ± 6	25 ± 8	21 ± 7	24 ± 5	24 ± 6
d18:1 (So)	49,651 ± 13,923	50,648 ± 13,053	37,057 ± 9774	60,817 ± 34,705	51,751 ± 13,794
d18:0 (Sa) *	2313 ± 524 ^A^	10,160 ± 4784 ^B^	3149 ± 1191 ^A^	2944 ± 1075 ^A^	9412 ± 2650 ^B^
GluSo *	321 ± 92 ^AB^	301 ± 61 ^AB^	432 ± 127 ^C^	250 ± 131 ^A^	372 ± 82 ^BC^
LysoSM	123 ± 25	119 ± 20	137 ± 26	100 ± 44	135 ± 31
Ceramides ^3^					
18:1/2:0	126 ± 25	138 ± 42	122 ± 24	134 ± 39	168 ± 48
18:1/14:0 *	2404 ± 491 ^A^	1472 ± 396 ^B^	2314 ± 660 ^A^	2402 ± 419 ^A^	1536 ± 405 ^B^
18:1/16:0 *	821,967 ± 125,521 ^A^	549,109 ± 70,814 ^B^	819,315 ± 151,516 ^A^	864,008 ± 109,172 ^A^	621,920 ± 116,635 ^B^
18:1/18:1 *	568 ± 163 ^A^	762 ± 144 ^A^	578 ± 190 ^A^	711 ± 311 ^A^	1017 ± 317 ^B^
18:1/18:0	89,899 ± 16,997	95,785 ± 16,495	95,023 ± 20,222	96,228 ± 19,789	110,841 ± 30,488
18:1/20:0	76,599 ± 15,945	78,651 ± 21,306	74,291 ± 21,561	66,223 ± 17,414	75,575 ± 16,013
18:1/22:2	6468 ± 1327	7146 ± 2463	5883 ± 1546	6164 ± 1752	5694 ± 1260
18:1/22:1	8944 ± 2055	8681 ± 2202	8000 ± 1884	7466 ± 1618	7928 ± 1651
18:1/22:0	276,112 ± 56,500	322,875 ± 74,191	274,742 ± 65,610	256,810 ± 72,396	324,251 ± 65,039
18:1/:23:1	7601 ± 1404	7372 ± 1550	6337 ± 1494	6680 ± 1166	6972 ± 1494
18:1/23:0 *	149,491 ± 15,447 ^AB^	175,097 ± 26,545 ^BC^	136,275 ± 28,192 ^A^	144,874 ± 31,886 ^A^	184,736 ± 3559 ^C^
18:1/24:2	270,944 ± 56,818	296,481 ± 79,986	250,626 ± 55,473	242,898 ± 55,874	255,533 ± 53,678
18:1/24:1	110,845 ± 21,795	113,820 ± 23,573	99,647 ± 19,098	96,379 ± 20,166	106,094 ± 17,652
18:1/24:0 *	106,041 ± 14,237 ^AB^	121,664 ± 20,682 ^BC^	102,467 ± 15,256 ^A^	103,407 ± 22,040 ^A^	129,104 ± 18,446 ^C^
18:1/25:1 *	4810 ± 787 ^AB^	5554 ± 1021 ^A^	4178 ± 939 ^B^	4465 ± 829 ^B^	5506 ± 914 ^A^
18:1/26:2	1675 ± 512	1793 ± 405	1491 ± 288	1645 ± 314	1685 ± 375
18:1/26:1	1024 ± 321	1048 ± 249	861 ± 179	991 ± 213	1093 ± 212
18:1/26:0	530 ± 50	517 ± 53	498 ± 39	546 ± 77	536 ± 20
Dihydroceramides					
18:0/2:0 *	28 ± 5 ^A^	70 ± 37 ^B^	23 ± 6 ^A^	27 ± 3 ^A^	55 ± 16 ^B^
18:0/16:0 *	39,330 ± 7838 ^AB^	49,331 ± 12,089 ^A^	34,423 ± 9441 ^B^	48,595 ± 13,933 ^A^	45,982 ± 11,738 ^A^
18:0/18:0 *	1408 ± 275 ^A^	2189 ± 468 ^B^	1252 ± 328 ^A^	1564 ± 406 ^A^	2268 ± 605 ^B^
18:0/20:0 *	779 ± 131 ^A^	867 ± 204 ^A^	569 ± 149 ^B^	704 ± 264 ^AB^	781 ± 199 ^A^
18:0/22:0 *	1544 ± 298 ^A^	2515 ± 457 ^B^	1166 ± 246 ^A^	1427 ± 425 ^A^	2338 ± 531 ^B^
18:0/23:0 *	211 ± 36 ^AB^	477 ± 110 ^C^	146 ± 41 ^B^	222 ± 63 ^A^	447 ± 94 ^C^
18:0/24:0 *	1919 ± 355 ^A^	2991 ± 477 ^B^	1473 ± 212 ^C^	1938 ± 457 ^A^	2924 ± 446 ^B^
Glycosylceramides ^4^					
Glu18:1/16:0 *	15,119 ± 3924 ^A^	5972 ± 2564 ^B^	14,059 ± 4659 ^A^	16,730 ± 4615 ^A^	7526 ± 2620 ^B^
Hex18:1/18:0	2339 ± 627	2347 ± 841	2431 ± 948	2881 ± 1055	3094 ± 1137
Hex18:1/22:0	15,832 ± 2737 ^A^	17,094 ± 5220 ^A^	16,269 ± 3605 ^A^	17,363 ± 4219 ^A^	21,613 ± 5417 ^B^
Glu18:1/24:1	3146 ± 736	3146 ± 1232	2819 ± 939	3282 ± 896	3629 ± 1002
Hex18:1/24:0 *	20,973 ± 5608 ^A^	27,628 ± 9389 ^A^	21,214 ± 9553 ^A^	25,339 ± 9306 ^A^	37,192 ± 10,065 ^B^
Lac18:1/16:0 *	6602 ± 2207 ^AB^	3483 ± 1155 ^C^	6169 ± 3735 ^ABC^	7637 ± 4290 ^A^	4166 ± 1665 ^BC^
Lac18:1/18:0	695 ± 210	693 ± 198	734 ±213	581 ± 160	841 ± 282
Sphingomyelins					
SM18:1/14:0	212 ± 133	133 ± 105	193 ± 97	147 ± 66	128 ± 53
SM18:1/16:0 *	109,485 ± 40,754 ^A^	70,543 ± 26,689 ^C^	106,364 ± 29,354 ^AB^	100,271 ± 31,924 ^AB^	77,609 ± 23,466 ^BC^
SM18:1/18:0	69,213 ± 35,767	90,806 ± 35,616	78,152 ± 13,186	66,340 ± 24,894	102,553 ± 45,202
SM18:1/20:0 *	25,422 ± 10,726 ^A^	36,254 ± 11,172 ^BC^	28,016 ± 4890 ^AB^	25,372 ± 7535 ^A^	40,712 ± 15,095 ^C^
SM18:1/22:2	476 ± 174	558 ± 223	496 ± 187	486 ± 178	548 ± 226
SM18:1/22:1	697 ± 224	772 ± 316	681 ± 185	647 ± 183	790 ± 234
SM18:1/22:0 *	264,502 ± 94,272 ^A^	435,724 ± 138,805 ^B^	295,101 ± 65,768 ^A^	281,556 ± 87,361 ^A^	460,163 ± 117,685 ^B^
SM18:1/23:1	522 ± 291	701 ± 403	479 ± 129	594 ± 223	655 ± 307
SM18:1/23:0 *	43,430 ± 14,488 ^A^	80,456 ± 31,499 ^B^	46,794 ± 13,648 ^A^	46,942 ± 16,591 ^A^	83,112 ± 19,614 ^B^
SM18:1/24:3	722 ± 465	648 ± 318	655 ± 293	734 ± 554	601 ± 220
SM18:1/24:2	13,038 ± 6339	17,281 ± 10,583	13,709 ± 7828	12,631 ± 7168	17,711 ± 8021
SM18:1/24:1	20,481 ± 8119	24,271 ± 14,149	19,771 ± 8159	18,890 ± 8496	25,992 ± 9622
SM18:1/24:0 *	42,082 ± 13,575 ^A^	73,438 ± 30,144 ^B^	47,161 ± 17,575 ^A^	45,702 ± 15,073 ^A^	79,637 ± 19,888 ^B^
SM18:1/25:2	522 ± 144	676 ± 235	558 ± 154	528 ± 140	646 ± 137
SM18:1/25:1	702 ± 419	1052 ± 800	680 ± 325	856 ± 384	973 ± 539
SM18:1/25:0 *	1755 ± 511 ^A^	2976 ± 1395 ^B^	1946 ± 791 ^A^	1955 ± 620 ^A^	3063 ± 782 ^B^
SM18:1/26:3	296 ± 87	276 ± 31	274 ± 24	264 ± 26	265 ± 18
SM18:1/26:2	532 ± 125	650 ± 206	566 ± 122	534 ± 126	626 ± 133
SM18:1/26:1	501 ± 105	578 ± 196	506 ± 108	496 ± 97	588 ± 122
SM18:1/26:0	663 ± 162	863 ± 286	738 ± 200	715 ± 155	925 ± 222
Dihydrosphingomyelins					
SM18:0/16:0 *	23,217 ± 7631 ^AB^	33,726 ± 11,917 ^BC^	24,714 ± 5839 ^AB^	21,586 ± 6263 ^A^	30,914 ± 7038 ^C^
SM18:0/18:0	796 ± 858	1693 ± 1700	912 ± 982	1217 ± 1251	1302 ± 1971
SM18:0/20:0 *	2655 ± 2024 ^AB^	4816 ± 3387 ^BC^	2052 ± 1469 ^A^	3174 ± 1731 ^ABC^	5238 ± 3062 ^C^
SM18:0/22:0 *	3633 ± 1124 ^A^	8095 ± 2592 ^B^	3531 ± 783 ^A^	3715 ± 989 ^A^	7563 ± 1847 ^B^
SM18:0/23:0 *	1314 ± 296 ^A^	3317 ± 1014 ^B^	1271 ± 316 ^A^	1416 ± 393 ^A^	3035 ± 632 ^B^
SM18:0/24:1 *	553 ± 117 ^A^	853 ± 403 ^B^	528 ± 127 ^A^	526 ± 106 ^A^	788 ± 171 ^B^
SM18:0/24:0 *	1368 ± 374 ^A^	2719 ± 1007 ^B^	1283 ± 400 ^A^	1447 ± 396 ^A^	2765 ± 736 ^B^

^1^ Turkeys were fed from the 55th to the 70th day of age with a mycotoxin-free control diet (CON) and different diets containing fumonisins (FB) at 20.2 mg/kg expressed as the sum of FB1 + FB2, deoxynivalenol (DON) at 5.12 mg/kg, zearalenone (ZEN) at 0.47 mg/kg and fumonisins, deoxynivalenol, and zearalenone (FDZ) at respective concentrations of 25.7, 5.15, and 0.57 mg/kg of feed. Results are corrected by the recovery measured for the IS as described in the materials and methods section and are expressed in pmol/g of the liver as means ± SD, n = 10 for each analyte per group, except the CON group where n = 8. ^2^ Deoxy-sphingosine, lactosyl-sphingosine, sphingosine 1-phosphate, and sphinganine 1-phosphate were only detected at the trace level in this study. dSa = deoxy-sphinganine; So = sphingosine; Sa = sphinganine; GluSo = glucosyl-sphingosine; LysoSM = lyso-sphingomyelin. ^3^ Ceramide 1-phosphate was only detected at the trace level in this study. ^4^ Glu = glucosyl; Hex = hexosyl; Lac = lactosyl. * Significant difference among groups (ANOVA, *p* < 0.05). Different letters in the same row indicate statistically different means (Duncan, *p* < 0.05).

**Table 2 ijms-23-02512-t002:** Correlation between some ceramides measured in the liver of turkeys fed different diets containing fusariotoxins ^1^.

Variables	18:1/14:0	18:1/16:0	18:1/18:0	18:1/20:0	18:1/22:0	18:1/23:0	18:1/24:0
18:1/14:0	1	**0.827**	**0.315**	0.076	−0.021	−0.035	0.011
18:1/16:0	**0.827**	**1**	**0.353**	0.145	0.065	0.054	0.063
18:1/18:0	**0.315**	**0.353**	**1**	**0.526**	**0.655**	**0.668**	**0.705**
18:1/20:0	0.076	0.145	**0.526**	**1**	**0.913**	**0.630**	**0.647**
18:1/22:0	−0.021	0.065	**0.655**	**0.913**	**1**	**0.856**	**0.869**
18:1/23:0	−0.035	0.054	**0.668**	**0.630**	**0.856**	**1**	**0.926**
18:1/24:0	0.011	0.063	**0.705**	**0.647**	**0.869**	**0.926**	**1**

^1^ Turkeys were fed from the 55th to the 70th day of age with a mycotoxin-free control diet (CON) and different diets containing fumonisins (FB) 20.2 mg/kg expressed as the sum of FB1 + FB2, deoxynivalenol (DON) at 5.12 mg/kg, zearalenone (ZEN) at 0.47 mg/kg, and fumonisins, deoxynivalenol, and zearalenone (FDZ) at respective concentrations of 25.7, 5.15, and 0.57 mg/kg of feed. Significant correlations (Pearson, *p* < 0.05) are reported in bold.

**Table 3 ijms-23-02512-t003:** Correlation between dihydroceramides measured in the liver of turkeys fed different diets containing fusariotoxins ^1^.

Variables	18:0/16:0	18:0/18:0	18:0/20:0	18:0/22:0	18:0/23:0	18:0/24:0
18:0/16:0	**1**	**0.742**	**0.480**	**0.565**	**0.614**	**0.608**
18:0/18:0	**0.742**	**1**	**0.589**	**0.869**	**0.899**	**0.902**
18:0/20:0	**0.480**	**0.589**	**1**	**0.775**	**0.538**	**0.580**
18:0/22:0	**0.565**	**0.869**	**0.775**	**1**	**0.924**	**0.920**
18:0/23:0	**0.614**	**0.899**	**0.538**	**0.924**	**1**	**0.961**
18:0/24:0	**0.608**	**0.902**	**0.580**	**0.920**	**0.961**	**1**

^1^ Turkeys were fed from the 55th to the 70th day of age with a mycotoxin-free control diet (CON) and different diets containing fumonisins (FB) at 20.2 mg/kg expressed as the sum of FB1 + FB2, deoxynivalenol (DON) at 5.12 mg/kg, zearalenone (ZEN) at 0.47 mg/kg, and fumonisins, deoxynivalenol, and zearalenone (FDZ) at respective concentrations of 25.7, 5.15, and 0.57 mg/kg of feed. Significant correlations (Pearson, *p* < 0.05) are reported in bold.

**Table 4 ijms-23-02512-t004:** Concentrations of fumonisins measured in the liver of turkeys fed different diets containing fusariotoxins ^1^.

	CON	FB	DON	ZEN	FDZ
FB1 ^2^	<LOQ	60.58 ± 18.18	<LOQ	<LOQ	60.89 ± 21.23
FB2 ^2^	<LOQ	5.81 ± 3.71	<LOQ	<LOQ	5.59 ± 2.18
FB3 ^2^	<LOQ	2.29 ± 0.86	<LOQ	<LOQ	2.56 ± 0.93

^1^ Turkeys were fed from the 55th to the 70th day of age with a mycotoxin-free control diet (CON) and different diets containing fumonisins (FB) 20.2 mg/kg expressed as the sum of FB1+FB2, deoxynivalenol (DON) 5.12 mg /kg, zearalenone (ZEN) 0.47 mg /kg and fumonisins, deoxynivalenol and zearalenone (FDZ) at respective concentrations of 25.7, 5.15 and 0.57 mg/kg feed. ^2^ Results are expressed in nmol/kg as mean ± SD, n = 10 for each analyte per group, except the CON group where n = 8. LOQ = limit of quantitation = 0.35 nmol/kg.

## Data Availability

Not applicable.

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
