# Peer review of "Targeted Analysis of Sphingolipids in Turkeys Fed Fusariotoxins: First Evidence of Key Changes That Could Help Explain Their Relative Resistance to Fumonisin Toxicity"

_ijms, 2022, doi:10.3390/ijms23052512_

Round 1

Reviewer 1 Report

Summary: The manuscript titled “Targeted Analysis of Sphingolipids in Turkeys Fed Fusariotoxins: First Evidence of Key Changes that Could Help Explain their Relative Resistance to Fumonisin Toxicity” is a study aimed to characterize variations in the concentrations of sphingolipids in the liver of turkeys fed fumonisins and identify changes in sphingolipids. The authors sought to determine whether these changes could explain the relative resistance of turkeys to fumonisins. In addition, they investigated the effects of deoxynivalenol and zearalenone on sphingolipids and characterized their interactions with fumonisins. To date, literature reports of similar studies using lipidomic approaches are scarce. Therefore, this is an exciting study with important information worth sharing with the scientific community.

The manuscript is well written; for the most part, some English language typos and errors are noted throughout the entire text. I would suggest professional editing to help clarify the language. Below are included some specific recommendations.

Specific comments and recommendations:

  1. Table 1 results: my recommendation is to convert this table to several bar graphs showing changes in different species of ceramides (panel A), DH-ceramides (panel B), sphingomyelins (panel C), glycosphingolipids (panel D), DH-SMs, etc. The current representation of data is tough to read and needs better visualization of the changes.
  2. Add a figure representing the overall quantitative changes in different SLs as a sum of all species.
  3. The numbers included in Table 1 are hard to understand and what they represent is unclear. Is this amount of lipid per mg of protein or something else? What normalization is used is also unclear. Please include this information in all figures and tables.

Author Response

Reviewer 1

Summary: The manuscript titled “Targeted Analysis of Sphingolipids in Turkeys Fed Fusariotoxins: First Evidence of Key Changes that Could Help Explain their Relative Resistance to Fumonisin Toxicity” is a study aimed to characterize variations in the concentrations of sphingolipids in the liver of turkeys fed fumonisins and identify changes in sphingolipids. The authors sought to determine whether these changes could explain the relative resistance of turkeys to fumonisins. In addition, they investigated the effects of deoxynivalenol and zearalenone on sphingolipids and characterized their interactions with fumonisins. To date, literature reports of similar studies using lipidomic approaches are scarce. Therefore, this is an exciting study with important information worth sharing with the scientific community.

The manuscript is well written; for the most part, some English language typos and errors are noted throughout the entire text. I would suggest professional editing to help clarify the language. Below are included some specific recommendations.

Thank you for your comments, your positive feedback and thank you for the time spent reviewing this work. This article was proofread before submission by a professional proofreader, we can provide the invoice if required. The typos that we were able to identify have been corrected in the revised version of the manuscript.

Specific comments and recommendations:

  1. Table 1 results: my recommendation is to convert this table to several bar graphs showing changes in different species of ceramides (panel A), DH-ceramides (panel B), sphingomyelins (panel C), glycosphingolipids (panel D), DH-SMs, etc. The current representation of data is tough to read and needs better visualization of the changes.

We agree that it is always difficult to choose between Tables and Figures as the best way to present the results. A tabular presentation was preferred for this work for the following reasons:

- the Table allows an easier comparison of the results between studies,

- the variations in concentrations between sphingolipids of the same class are often so great that it would have been difficult to group several compounds together in the same Figure while keeping the y-axis readable,

- the number of sphingolipids analyzed and the number of groups studied would have required either a very large number of figures or the creation of tiny figures whose y-axis would have been very difficult to read.

  1. Add a figure representing the overall quantitative changes in different SLs as a sum of all species.

Thank you for this suggestion. A Figure (Figure 2 in the new version of the manuscript) showing the concentrations in the different sphingolipid groups has been added at the beginning of the article. In connection with the previous comment, this Figure completes Table 1 and makes it possible to insist on the fact that an analysis conducted at the level of the sphingolipid class alone does not reveal all the alterations observed.

  1. The numbers included in Table 1 are hard to understand and what they represent is unclear. Is this amount of lipid per mg of protein or something else? What normalization is used is also unclear. Please include this information in all figures and tables.

Thank you for this observation. Sphingolipid concentrations were reported after correction by the recovery measured for each sample for the internal standard representative of each sphingolipid class and expressed in pmol/g of liver. This point has been clarified in Table 1 and the relevant Figures in the new version of the manuscript.

Reviewer 2 Report

This is an excellent manuscript and has numerous fine/new assumptions.

I am in a special position, since I am absolutely satisfied with the work and its presentation. Criticizing it would be not correct.

I have only 2-3 minor questions:

  1. please add some literature references to Fig 1
  2. Please add dimensions to Table 1 (I guess nmol/g)
  3. Please add some or at least one supportive source or explanation, why liver has been the mere target of the study, and why were other organs not studied.

Apart from these minor points:

  • the study is excellent
  • analytics are over-professional!
  • results' presentation is CLEAR
  • PLS-DA is absolutely finely applied and clearly interpreted

I absolutely clearly recommend publication in TOXINS, and just recommend my suggestions to be considered.

Author Response

Reviewer 2

This is an excellent manuscript and has numerous fine/new assumptions.

I am in a special position, since I am absolutely satisfied with the work and its presentation. Criticizing it would be not correct.

Thank you for your comments and for the time spent reviewing this work. Thank you also for your very positive feedback, it is always nice to know that the efforts made are appreciated.

I have only 2-3 minor questions:

  1. please add some literature references to Fig 1

Thank you for this remark, this point has been corrected in the revised version of the manuscript

  1. Please add dimensions to Table 1 (I guess nmol/g)

Thank you for this remark, the results are in pmol/g of liver, this point has been corrected in the revised version of the manuscript.

  1. Please add some or at least one supportive source or explanation, why liver has been the mere target of the study, and why were other organs not studied.

Thank you for your comment, the liver was studied first as it is known to be the main target organ of fumonisins in poultry. It was also easier to start with the liver as it is one of the few tissues for which there is some literature available to compare effects between studies. This point has been clarified at the beginning of the discussion of the revised version of the manuscript. In addition, work is underway to determine the effects of fumonisins on other organs, notably the kidneys, but also the lung and brain.

Apart from these minor points:

  • the study is excellent
  • analytics are over-professional!
  • results' presentation is CLEAR
  • PLS-DA is absolutely finely applied and clearly interpreted

I absolutely clearly recommend publication in TOXINS, and just recommend my suggestions to be considered.

Thanks again for you very positive feedback on the work done